# Short-term outcomes after emergency surgery for complicated peptic ulcer disease from the UK National Emergency Laparotomy Audit: a cohort study

Benjamin E Byrne,[1] Michael Bassett,[2,3] Chris A Rogers,[4] Iain D Anderson,[2,5] Ian Beckingham,[6,7] Jane M Blazeby,[1,8] the Association of Upper Gastrointestinal Surgeons for the National Emergency Laparotomy Project Team

Parts of this work have been presented at the Association of Surgeons of Great Britain and Ireland International Surgical Congress, Glasgow, May 2017, and at the Association of Upper Gastro-Intestinal Surgeons 20th Annual Scientific Meeting, Cork, September 2017.

For numbered affiliations see end of article.

**Correspondence to**
Benjamin E Byrne;
benbyrne@doctors.org.uk

## ABSTRACT

**Objectives** This study used national audit data to describe current management and outcomes of patients undergoing surgery for complications of peptic ulcer disease (PUD), including perforation and bleeding. It was also planned to explore factors associated with fatal outcome after surgery for perforated ulcers. These analyses were designed to provide a thorough understanding of current practice and identify potentially modifiable factors associated with outcome as targets for future quality improvement.

**Design** National cohort study using National Emergency Laparotomy Audit (NELA) data.

**Setting** English and Welsh hospitals within the National Health Service.

**Participants** Adult patients admitted as an emergency with perforated or bleeding PUD between December 2013 and November 2015.

**Interventions** Laparotomy for bleeding or perforated peptic ulcer.

**Primary and secondary outcome measures** The primary outcome was 60-day in-hospital mortality. Secondary outcomes included length of postoperative stay, readmission and reoperation rate.

**Results** 2444 and 382 procedures were performed for perforated and bleeding ulcers, respectively. In-hospital 60-day mortality rates were 287/2444 (11.7%, 95% CI 10.5% to 13.1%) for perforations, and 68/382 (17.8%, 95% CI 14.1% to 22.0%) for bleeding. Median (IQR) 2-year institutional volume was 12 (7–17) and 2 (1–3) for perforation and bleeding, respectively. In the exploratory analysis, age, American Society of Anesthesiology score and preoperative systolic blood pressure were associated with mortality, with no association with time from admission to operation, surgeon grade or operative approach.

**Conclusions** Patients undergoing surgery for complicated PUD face a high 60-day mortality risk. Exploratory analyses suggested fatal outcome was primarily associated with patient rather than provider care factors. Therefore, it may be challenging to reduce mortality rates further. NELA data provide important benchmarking for patient consent and has highlighted low institutional volume and high mortality

### Strengths and limitations of this study

► This multicentre study examined usual clinical practice across a large number of hospitals in the National Health Service in England and Wales, representing the largest study of complicated peptic ulcer disease yet reported in the UK.

► Structured data were collected prospectively, mitigating against bias associated with retrospective study design.

► However, case ascertainment within the entire National Emergency Laparotomy Audit patient cohort, the main data set from which the current study data were extracted, was 83% and missing data may have introduced unknown biases into the study.

rates after surgery for bleeding peptic ulcers as a target for future research and improvement.

## INTRODUCTION

Surgical treatment of peptic ulcer disease (PUD) has changed markedly over recent years. Overall operative intervention has declined, with a substantial fall in elective procedures such as gastric resection, vagotomy and pyloroplasty.[1–3] The role of surgery is now largely restricted to the emergency setting, for management of the complications of PUD.[4] Surgical repair is the treatment of choice for perforated PUD and is a second-line or third-line treatment for bleeding ulcers that cannot be managed by endoscopic and/or radiological means. Earlier studies and nationwide audits show postoperative mortality following emergency surgery for perforated or bleeding ulcers to range from 9.1% to 26.5% although data from contemporary UK practice are lacking.[2 5–8]

The National Emergency Laparotomy Audit (NELA) is a mandatory audit that captures rich data about the care of patients undergoing a range of emergency bowel operations in England and Wales. NELA was established in 2012 and is run by the Royal College of Anaesthetists, in collaboration with the Clinical Effectiveness Unit at the Royal College of Surgeons of England. The audit aims to improve the quality of care for patients undergoing emergency laparotomy, by collecting information on patients, the processes of care they receive and their short-term outcomes. These data are fed back locally, as well as being analysed nationally, and compared with accepted audit standards. To date, there have been three audit reports, most recently documenting a 30-day postoperative mortality rate of 10.6% (95% CI 10.2% to 11.0%) and a median length of stay of 11 days across the range of patients and conditions included.[9–11]

The present study aimed to use NELA data to identify patients undergoing surgery for perforated or bleeding PUD, to describe the latest management and short-term outcomes for these patients. The study also explored factors that may be associated with mortality after surgery for perforated PUD. A thorough appraisal of current practice is critical for benchmarking performance, and appropriately directing future research and quality improvement.

## METHODS

The NELA database contains information collected at the level of individual operations for patients, covering details of the admission, preoperative management and risk stratification, intraoperative details, postoperative risk and patient outcomes. From this database, patients aged 18 years or over undergoing 'Peptic ulcer—suture or repair of perforation' or 'Peptic ulcer—oversew of bleed' as their first, main surgical procedure after admission, between 1 December 2013 and 30 November 2015, were selected for inclusion. Reoperations and patients undergoing 'Gastric surgery—other', which is likely to have included formal surgical resection, were excluded. Data for the first and second years of the study were extracted on 1 February in 2015 and 2016, respectively. Data on age, sex, American Society of Anesthesiologists (ASA) score, preoperative heart rate (HR), preoperative systolic blood pressure (SBP), preoperative predicted mortality (Portsmouth-Physiological and Operative Severity Score for the enUmeration of Mortality and Morbidity (P-POSSUM))[12] and morbidity (POSSUM) were extracted.[13] NELA specifies recording of HR and SBP values closest to the time of booking the patient for theatre. Preoperative care details, including the use of CT, time from admission to operation, time from admission to decision to operate, time from decision to operate to operation and time from admission to antibiotics were also recorded. Information on the grade of most senior operating surgeon and surgical approach (open or minimal access), as well as intraoperative findings (extent and type of peritoneal

contamination, see online supplementary table 1), were examined, along with the immediate postoperative level of care (ward, level 2 or high dependency unit (HDU), and level 3 or intensive therapy unit (ITU)) and hospital procedure volume. The following outcomes were examined: total length of stay in an enhanced care setting (HDU or ITU); total postoperative length of stay; return to theatre; and in-hospital death within 60 days of the primary operative procedure.

## Statistical analysis

Patients were grouped and analysed separately according to presentation with perforation or bleeding secondary to PUD. Continuous data were described using mean and SD or median and IQR if skewed, and category data were summarised as number and percentage. Length of stay was summarised using survival methods with deaths prior to discharge treated as censored observations.

For the exploratory analysis of patients with perforated PUD, associations between patient, care and operative factors and mortality were examined using multilevel logistic regression, with hospital fitted as random effect. For this analysis alone, only patients undergoing surgery for perforation within 48 hours of admission to hospital were included. This was designed to exclude patients who developed a perforation during their admission, and patients undergoing surgery after an unsuccessful period of non-operative management. Such patients may represent a different population with a different risk profile. For example, severely comorbid patients with mild clinical signs may preferentially be selected for initial non-operative management. The following variables were selected for exploration by consensus within the working group before analysis: age, sex, ASA, HR, SBP, preoperative CT, time from admission to operation, grade of senior operating surgeon, operative approach, peritoneal contamination type, peritoneal contamination extent and postoperative care level. Variables with many categories were grouped for analysis (see online supplementary table 1). Fractional polynomials were used to describe the relationship between continuous variables and mortality (online supplementary table 2). Data that were clearly incorrect were recoded as missing. The analysis was restricted to cases with complete data for the variables of interest. All variables were included in the model and not selected based on statistical significance. All analyses were conducted using Stata 14.0 (StataCorp, College Station, Texas, USA).

## Ethics

NELA has approval from the Health Research Authority's Confidentiality Advisory Group for 'Use of Patient Identifiable Information without Consent' (Section 251 of National Health Service (NHS) Act 2006 and Health Service (Control of Patient Information) Regulations 2002). The present analysis was performed under NELA's remit to understand and inform the delivery of care to patients undergoing emergency laparotomy. The data

extract included anonymised patient-level data. Therefore, further approval by a research ethics committee was not required. Participating Trusts follow local governance arrangements for audit registration. Patient data are uploaded via an encrypted website to a secure server. Access is carefully restricted and data used in accordance with the Caldicott principles.[14]

## Patient involvement
Patients were not involved in any aspect of the design or conduct of this study.

## RESULTS
### Patient characteristics and hospital volume
During the study period, 43 321 emergency laparotomies were identified at 192 hospitals in England and Wales. Data on 2444 (5.5%) perforated peptic ulcers and 382 (0.9%) bleeding ulcers were retrieved from 186 (96.9%) contributing hospitals. Patient characteristics are shown in table 1. Over the 2-year period, the median number of cases per hospital was 12 (IQR 7–17) and 2 (IQR 1–3) for perforated and bleeding ulcers, respectively.

### Preoperative care
Preoperative imaging differed according to diagnosis, with the majority (1792/2444, 73.3%) of patients undergoing treatment for perforated ulcers receiving a preoperative CT scan (table 2), compared with 101/382 (26.4%) of patients with a bleeding ulcer.

Median interval from admission to surgery was 8.8 hours (IQR 5.3–18.9) in patients with a perforated ulcer, and 30.4 hours (IQR 9.4–107.8) in those with a bleeding ulcer. Median interval from decision to operate to surgery was 2.0 (IQR 1.2–3.4) and 1.1 (IQR 0.5–2.1) hours in patients with perforations and bleeding ulcers, respectively. Patients admitted with a perforated ulcer received their first dose of antibiotics at a median of 4.6 (IQR 2.1–10.1; data recorded for 2085/2444 (85.3%) of patients) hours after admission, and those undergoing surgery for bleeding ulcers received antibiotics at a median of 11.8 hours (IQR 3.8–47.8; data for 273/382 (71.5%) of patients).

### Operative details and postoperative care level
Consultants were recorded as the senior surgeon in the majority of cases and most surgeries were performed via the open approach (table 3). However, 489/2 444 (20.0%) patients underwent some form of laparoscopic surgery for their perforated ulcer, with 320 (13.1%) procedures completed laparoscopically. The nature and extent of peritoneal contamination differed according to the clinical problem. The majority of patients with perforated ulcers had significant contamination affecting multiple quadrants.

Most patients who underwent surgery to repair a perforated ulcer (1426/2444, 58.3%) were transferred to an HDU or ITU environment, with the remainder being transferred to a ward. A much higher proportion of patients operated on for bleeding went to HDU or ITU after surgery (311/382, 81.4%).

## Outcomes
Among patients transferred to HDU or ITU, the median postoperative stay in an enhanced care environment was 4 days for both groups (table 4). The median total postoperative stay was 8.4 days for patients treated for a perforated ulcer, compared with a longer median stay of 15.0 days for bleeding ulcers. The rate of return to theatre was lower among patients operated on for perforation (136/2 444, 5.6%) than after surgery for bleeding (36/382, 9.4%). In each group, three patients died in theatre. The overall, 60-day in-hospital mortality was 287/2 444 (11.7%, 95% CI 10.5% to 13.1%) after surgery for a perforated ulcer, and 68/382 (17.8%, 95% CI 14.1% to 22.0%) after oversew of a bleeding ulcer.

## Exploratory analysis
Of 2327 patients where time from admission to operation was recorded, 2231 (96.1%) underwent surgery for perforation in the first 48 hours after admission. Complete data for regression analysis were available for 2162 (96.7%) of these patients. Variables identified as significantly associated with in-hospital 60-day mortality (after accounting for other variables) were age, ASA, preoperative SBP and postoperative care level (table 5). For each increasing year of age, the risk of death rose by 5.0% (95% CI 3.5% to 6.5%), meaning that an increase of 10 years of age was associated with increased risk of death of 63.3% (95% CI 41.6% to 88.4%). An ASA score of 4 or 5 was associated with a markedly elevated risk of fatal outcome compared with an ASA of 1. There was also an increased risk of death for patients going to HDU or ITU compared with those transferred directly to a ward. The association between preoperative SBP and postoperative mortality was non-linear (illustrated in figure 1). Extremes of low or high SBP were associated with increased risk of death. There was no statistically significant association observed between patient sex, preoperative HR, use of preoperative CT, time from admission to operation, operating surgeon, operative approach, intraoperative contamination type or extent and subsequent in-hospital mortality.

## DISCUSSION
This is the first national study in the UK of complicated PUD requiring emergency surgery. Using 2 years of NELA data, we identified 2444 and 382 patients from 186 English and Welsh hospitals undergoing surgery for perforated or bleeding PUD. The postoperative in-hospital 60 day mortality rates were 11.7% and 17.8%, respectively. In exploratory analysis, mortality after repair of perforated ulcer was primarily associated with patient factors, rather than potentially modifiable aspects of the care provided. This may make it difficult to reduce mortality rates further. Average institutional surgical volume for bleeding ulcers

**Table 1** Preoperative details of patients undergoing surgery for perforation or bleeding

|  | Perforation | Bleed | All PUD |
|---|---|---|---|
|  | n=2444 (%) | n=382 (%) | n=2826 (%) |
| **Age in years (mean (SD))** |  |  |  |
| Mean | 57.8 (19.4) | 65.0 (16.3) | 58.8 (19.2) |
| **Sex** |  |  |  |
| Male | 1450 (59.3) | 240 (62.8) | 1690 (59.8) |
| **ASA** |  |  |  |
| 1 | 569 (23.3) | 30 (7.9) | 599 (21.2) |
| 2 | 738 (30.2) | 64 (16.8) | 802 (28.4) |
| 3 | 611 (25.0) | 98 (25.7) | 709 (25.1) |
| 4 | 461 (18.9) | 158 (41.4) | 619 (21.9) |
| 5 | 65 (2.7) | 32 (8.4) | 97 (3.4) |
| **Preoperative heart rate** |  |  |  |
| <80 | 449 (18.6) | 47 (12.5) | 496 (17.8) |
| 80–99 | 928 (38.4) | 140 (37.1) | 1068 (38.2) |
| 100–119 | 704 (29.1) | 109 (28.9) | 813 (29.1) |
| 120–139 | 267 (11.0) | 65 (17.2) | 332 (11.9) |
| ≥140 | 69 (2.9) | 16 (4.2) | 85 (3.0) |
| **Preoperative systolic blood pressure** |  |  |  |
| <80 | 63 (2.6) | 37 (11.6) | 100 (3.6) |
| 80–99 | 260 (10.8) | 96 (30.1) | 356 (12.8) |
| 100–119 | 670 (27.8) | 118 (37.0) | 788 (28.3) |
| 120–139 | 831 (34.5) | 68 (21.3) | 899 (32.3) |
| 140–159 | 429 (17.8) | 43 (11.4) | 472 (16.9) |
| ≥160 | 157 (6.5) | 15 (4.0) | 172 (6.2) |
| **Predicted mortality (P-POSSUM)** |  |  |  |
| <5% | 935 (38.3) | 49 (12.8) | 984 (34.8) |
| 5%–9% | 416 (17.0) | 37 (9.7) | 453 (16.0) |
| 10%–24% | 445 (18.2) | 74 (19.4) | 519 (18.4) |
| 25%–49% | 292 (11.9) | 83 (21.7) | 375 (13.3) |
| ≥50% | 356 (14.6) | 139 (36.4) | 495 (17.5) |
| **Predicted morbidity (POSSUM)** |  |  |  |
| <25% | 54 (2.2) | 2 (0.5) | 56 (2.0) |
| 25%–49% | 385 (15.8) | 16 (4.2) | 401 (14.2) |
| 50%–74% | 747 (30.6) | 53 (13.9) | 800 (28.3) |
| ≥75% | 1258 (51.5) | 311 (81.4) | 1569 (55.5) |

ASA, American Society of Anesthesiology score; POSSUM, Physiological and Operative Severity Score for the Enumeration of Mortality and Morbidity; P-POSSUM, Portsmouth-POSSUM; PUD, peptic ulcer disease.

was very low, and these patients had the highest mortality risk, highlighting the challenge and urgent need for further work to understand and improve outcomes in this group.

While mortality rates among the included patients were high, the reported results compare favourably with data from other research. Recent European studies have reported 90-day mortality rates from 19.2% to 29.8% after surgery for perforated PUD.[5 15] Among patients undergoing surgery for bleeding peptic ulcers, 30-day mortality rates were higher, ranging from 23.7% to 25.6%,[7 16] with previous UK research revealing a 30% (95% CI 22% to 38%) postoperative mortality rate.[17] A recent large US study using American College of Surgeons National Surgical Quality Improvement Programme data demonstrated similar rates to those observed in this study with 12.1% (95% CI 10.8% to 13.5%) and 18.6% (95% CI 15.9% to 21.5%) 30-day

**Table 2** Details of preoperative care of patients undergoing surgery for perforation or bleeding

| | Perforation | Bleed | All PUD |
|---|---|---|---|
| | n=2444 (%) | n=382 (%) | n=2826 (%) |
| Preoperative CT | | | |
| Yes | 1792 (74.1) | 101 (26.8) | 1893 (67.7) |
| No | 626 (25.9) | 276 (73.2) | 902 (32.3) |
| Time in hours (median (IQR)) | | | |
| Admission to operation | 8.8 (5.3–18.9) | 30.4 (9.4–107.8) | 9.7 (5.5–23.4) |
| Admission to decision to operate | 6.0 (3.1–14.6) | 29.3 (7.5–119.3) | 6.5 (3.3–19.4) |
| Decision to operate to operation | 2.0 (1.2–3.4) | 1.1 (0.5–2.1) | 1.9 (1.1–3.2) |
| Admission to first antibiotics | 4.6 (2.1–10.1) | 11.8 (3.8–47.8) | 5.0 (2.3–11.9) |

PUD, peptic ulcer disease.

mortality after surgery for perforation and bleeding PUD, respectively.

The exploratory analysis of factors associated with mortality after repair of perforation generated new, unexpected findings. The significant associations between age, ASA and preoperative SBP and postoperative mortality are unsurprising and consistent with previous research and risk prediction models.[18–20] Accurate and reproducible risk prediction to guide individual patient care would be useful but has been proven difficult. In this manuscript, our exploratory analysis was principally concerned with a broad assessment of current practice and care provision. We were surprised to find a lack of association between time from admission to surgery and mortality, and this disagrees with the published literature. For example, Buck et al reported that after adjusting for prognostic variables, each hour of surgical delay during the first 24 hours of admission was associated with a 2.4% (95% CI 1.1% to 3.7%) decrease in the probability of survival.[21] However, that study used less nuanced modelling of continuous variables such as age or shock, which were reduced to dichotomous variables. In addition, they did not describe any exclusion criteria based on time from admission to surgery, raising the possibility

**Table 3** Operative details and postoperative destination for patients undergoing surgery for perforation or bleeding

| | Perforation | Bleed | All PUD |
|---|---|---|---|
| | n=2444 (%) | n=382 (%) | n=2826 (%) |
| Operation | | | |
| Senior surgeon | | | |
| Consultant | 1763 (72.1) | 347 (90.1) | 2110 (74.7) |
| Specialty trainee | 453 (18.5) | 27 (7.1) | 480 (17.0) |
| Other | 228 (9.3) | 8 (2.1) | 236 (8.4) |
| Approach | | | |
| Open | 1955 (80.0) | 367 (96.1) | 2322 (82.2) |
| Laparoscopic (including assisted) | 320 (13.1) | 10 (2.6) | 330 (11.7) |
| Laparoscopic converted | 169 (6.9) | 5 (1.3) | 174 (6.2) |
| Contamination type | | | |
| None/minimal | 425 (17.4) | 250 (65.4) | 675 (23.9) |
| Significant | 2019 (82.6) | 132 (34.6) | 2151 (76.1) |
| Contamination extent | | | |
| None/single quadrant | 753 (30.8) | 319 (83.5) | 1072 (37.9) |
| Multiple quadrants | 1691 (69.2) | 63 (16.5) | 1754 (62.1) |
| Postoperative care level | | | |
| Ward (level 1) | 1015 (41.5) | 68 (17.8) | 1083 (38.4) |
| HDU (level 2) | 652 (26.7) | 99 (25.9) | 751 (26.6) |
| ITU (level 3) | 774 (31.7) | 212 (55.5) | 986 (35.0) |

HDU, high dependency unit; ITU, intensive therapy unit; PUD, peptic ulcer disease.

**Table 4** Outcomes of patients undergoing surgery for perforation or bleeding.

| | Perforation | Bleed | All PUD |
|---|---|---|---|
| | n=2444 (%) | n=382 (%) | n=2826 (%) |
| Length of stay (days) Median (IQR) | | | |
| HDU/ITU | 4.0 (2.0–7.0) | 4.0 (2.0–8.0) | 4.0 (2.0–7.0) |
| Total | 8.4 (5.2–18.4) | 15.0 (7.5–29.0) | 9.2 (5.4–20.0) |
| Return to theatre | 136 (5.6) | 36 (9.4) | 172 (6.1) |
| Mortality in-hospital within 60 days | 287 (11.7) | 68 (17.8) | 355 (12.6) |
| (Died in theatre) | 3 (0.1) | 3 (0.8) | 6 (0.2) |

HDU, high dependency unit; ITU, intensive therapy unit; PUD, peptic ulcer disease.

**Table 5** Multilevel logistic regression results, examining factors associated with 60-day in-hospital mortality after surgery for perforation

| | | 95% CI | | |
|---|---|---|---|---|
| | OR | Lower | Upper | P values |
| Age (per year) | 1.05 | 1.04 | 1.07 | <0.001 |
| Sex | | | | |
| Male | 1.00 | | | |
| Female | 1.00 | 0.70 | 1.42 | 0.999 |
| ASA | | | | |
| 1 | 1.00 | | | <0.001 |
| 2 | 0.77 | 0.26 | 2.26 | |
| 3 | 2.10 | 0.77 | 5.76 | |
| 4 & 5 | 7.19 | 2.62 | 19.73 | |
| Preoperative heart rate (per 10 bpm) | 1.03 | 0.95 | 1.11 | 0.529 |
| Preoperative systolic blood pressure* | | | | <0.001 |
| Preoperative CT | | | | |
| No | 1.00 | | | |
| Yes | 1.41 | 0.90 | 2.22 | 0.133 |
| Time from admission to operation (per hour) | 1.01 | 0.99 | 1.02 | 0.392 |
| Operating surgeon | | | | |
| Consultant | 1.00 | | | |
| Non-consultant | 0.90 | 0.57 | 1.40 | 0.633 |
| Operative approach | | | | |
| Open | 1.00 | | | |
| Laparoscopic (inc. assisted) | 0.78 | 0.40 | 1.50 | 0.459 |
| Contamination type | | | | |
| None/minimal | 1.00 | | | |
| Significant | 0.88 | 0.49 | 1.58 | 0.660 |
| Contamination extent | | | | |
| None/single quadrant | 1.00 | | | |
| Multiple quadrants | 1.14 | 0.70 | 1.84 | 0.605 |
| Postoperative destination | | | | |
| Ward | 1.00 | | | |
| HDU or ITU | 2.22 | 1.20 | 4.11 | 0.011 |

Analysis restricted to patients undergoing surgery within 48 hours of admission.
*Non-linear relationship.
ASA, American Society of Anesthesiology score; HDU, high dependency unit; ITU, intensive therapy unit.

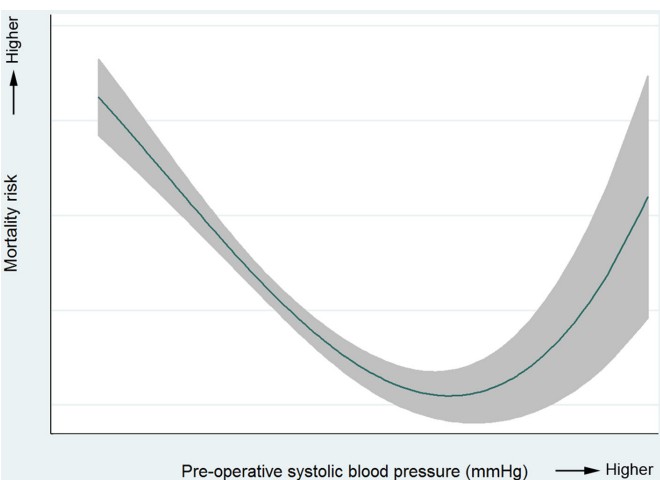

**Figure 1** Illustration of non-linear relationship between preoperative systolic blood pressure and 60-day in-hospital mortality for patients undergoing surgery for perforation only.

that their cohort included patients undergoing surgery after failed conservative management, and those developing a perforation during their hospital admission. The present lack of association between time from admission to operation may have important clinical implications, as it suggests that focusing efforts on reducing the interval from admission to operation may not be the best way to reduce mortality rates. However, preoperative blood pressure was associated with subsequent mortality, and this is a potentially modifiable variable. Future research and quality improvement should evaluate the role of preoperative optimisation, at the cost of a short delay in transfer to the operating theatre, as a possible strategy to improve outcomes for patients who have already experienced a perforation.

This study found that 13.1% of patients underwent surgical repair of their perforated ulcer via a laparoscopic approach. A further 6.9% were converted from laparoscopic to open. Although the reasons for conversion were not recorded, it is possible that some patients underwent an initial diagnostic laparoscopy before proceeding directly to open repair once the diagnosis was made. A smaller Danish study of 726 patients undergoing surgery for perforated PUD reported a laparoscopy rate of 32.8%, with 24.5% converted from laparoscopic to open.[22] The lower rate in the present study may represent under-reporting, as, anecdotally, some clinicians may not have considered a laparoscopic suture repair eligible for a laparotomy audit. Future comparison with Hospital Episode Statistics administrative data, available for all NHS activity, could test this hypothesis. Alternatively, the lower rate may be a true reflection of practice, and lack of skills or confidence in performing laparoscopic repair. The lack of association between operative approach and outcome may suggest that patients are being treated laparoscopically on the availability of appropriately skilled surgeons, rather than through careful case selection or 'picking winners'. Further and more detailed analysis is warranted.

This study has not defined clear ways to improve the survival of the patients included. It has, however, identified aspects of the care provided that were not associated with mortality, suggesting that these should not be the primary focus for immediate quality improvement. The results provide no evidence that more rapid transfer to the operating theatre, greater consultant presence or adoption of minimal access techniques would improve survival rates. However, preoperative SBP may be an appropriate target for future research and quality improvement. Selection of patients for postoperative care in the HDU or ITU environment, and other variables not analysed due to missing data, such as time from admission to antibiotics, should also be investigated further.

The results highlight the low institutional volume for the included procedures, particularly for bleeding PUD requiring surgery. Endoscopy is the first-line investigation and treatment for upper gastrointestinal bleeding, and previous large studies have demonstrated the high success rates of endoscopic therapy for bleeding ulcers, with surgery required in 1.9%–5.4% of cases.[5 16 17] National guidance in the UK suggests interventional radiology and embolisation should be offered as second-line treatment,[23] but few hospitals have 24/7 access to this service. In 2014, 45% of services in England did not have access to either local or networked interventional radiology out of hours.[24] It is likely that many of the 101 of 382 (26.8%) patients with bleeding ulcers that underwent CT had CT angiograms, though the specific details or preoperative CT are not collected in NELA. However, this information would be useful in future updates to the NELA data template. When requiring surgery, patients with bleeding ulcers are high risk, as reflected in their ASA scores, with associated high levels of senior involvement in theatre. Low procedure volumes make it difficult to develop expertise managing these patients, which in turn may make it difficult to improve outcomes. In several surgical specialties, higher procedure volume has been associated with improved outcomes.[25–27] However, it is not clear whether such a volume–outcome relationship exists for emergency surgery for bleeding peptic ulcers and it may be difficult to centralise secondary treatment required for an unstable patient. Further research, using quantitative databases and case studies in different centres, may determine future strategies to improve care for these patients.

This study has several strengths, as a nationwide prospective audit. However, there are limitations. While participation is mandatory, case ascertainment was estimated at 83% in the first 2 years of the audit.[9 10] No eligible procedures were identified in six (3.1%) of hospitals participating in NELA. Patients with complicated PUD that were successfully managed without surgery are not included in NELA. Research in other areas has found that voluntary clinical databases typically demonstrate a lower mortality rate than population-based administrative data.[28 29] This may represent selection bias and it is possible that mortality rates across the country are higher than observed, further

highlighting the need for more work in this area. In addition, deaths after discharge, or during the index admission but more than 60 days after the operative procedure, were not included. Another limitation is possible variation in coding of information, which depends on how different observers interpret the terms. For example, coding of contamination in bleeding ulcers may have reflected existing contamination, or it may have reflected contamination due to the enterotomy required to visualise and treat the bleeding ulcer. It is not possible to retrospectively check the accuracy of such data, which must be taken at face value. While data completeness was satisfactory for the analysis presented, it is not possible to determine whether missing data introduced systematic bias. The extent of missing data precluded exploratory analysis of further variables of interest, such as time from admission to antibiotics, and time from admission to decision to operate. While the results may be cautiously generalised to similar populations and healthcare systems, differences in care organisation may limit broad applicability.

In summary, this national study has demonstrated mortality rates within the NHS in England and Wales that compare favourably with previously published international results. The overall rate of mortality, however, remains high. Exploratory analysis suggested fatal outcome after surgery for perforation was primarily associated with patient factors rather than the care provided, and this may make further improvement difficult. As NELA accrues more data over the remaining years of the project, it may be feasible to explore the association between other, modifiable care factors, such as time to antibiotics, and clinical outcomes and this could aid further research. Surgical management of bleeding PUD represents an area of practice with very low volume and high postoperative mortality that mandates further investigation. Centralisation may be considered, though this could be difficult due to the acuity of patients requiring surgery in this setting. Research using future audit data may guide quality improvement efforts, to benefit patients requiring surgery for complications of PUD.

**Author affiliations**
¹Centre for Surgical Research, Population Health Sciences, Bristol Medical School, University of Bristol, Bristol, UK
²National Emergency Laparotomy Audit, The Royal College of Anaesthetists, London, UK
³Department of Anaesthesia, Manchester University NHS Foundation Trust, Manchester, UK
⁴Clinical Trials and Evaluation Unit, Bristol Medical School, University of Bristol, Bristol, UK
⁵Department of Surgery, Salford Royal NHS Foundation Trust, Salford, UK
⁶Department of Surgery, Nottingham University Hospitals NHS Trust, Nottingham, UK
⁷Association of Upper Gastrointestinal Surgeons, Royal College of Surgeons of England, London, UK
⁸Department of Surgery, University Hospitals Bristol NHS Foundation Trust, Bristol, UK

**Acknowledgements** The authors would like to thank the members of the NELA Project Team during the data collection period of this study: Mr Martin Cripps, Professor David Cromwell, Dr Emma Davies, Ms Sharon Drake, Dr Mike Galsworthy, Professor Mike Grocott, Dr Angela Kuryba, Mr Jose Lourtie, Dr Ramani Moonesinghe, Dr Dave Murray, Dr Matt Oliver, Mr Dimitri Papadimitriou, Dr Carol Peden, Dr Kate Walker.

**Contributors** IDA, IB and JMB conceived the study. MB and IDA acquired the data. BEB, MB and CAR analysed the data. All authors interpreted the data. BEB drafted the manuscript. MB, CAR, IDA, IB and JMB critically revised the manuscript for important intellectual content. All authors approved the final version for publication.

**Funding** BEB is an academic clinical fellow in surgery supported by the National Institute for Health Research (NIHR). JMB is an NIHR Senior Investigator. This work was undertaken with the support of the MRC ConDuCT-II (Collaboration and innovation for Difficult and Complex randomised controlled Trials In Invasive procedures) Hub for Trials Methodology Research (MR/K025643/1). The study was also supported by the NIHR Biomedical Research Centre at the University Hospitals Bristol NHS Foundation Trust and the University of Bristol. The National Emergency Laparotomy Audit is commissioned by the Healthcare Quality Improvement Partnership as part of the National Clinical Audit Programme on behalf of NHS England and the Welsh Government.

**Disclaimer** The views expressed are those of the authors and not necessarily those of the UK National Health Service, National Institute for Health Research, or the Department of Health. The funders had no role in study design, data collection and analysis, decision to publish, or preparation of the manuscript.

**Competing interests** None declared.

**Patient consent** Not required.

**Provenance and peer review** Not commissioned; externally peer reviewed.

**Data sharing statement** No additional data available.

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
