## [Reviewer comments · BMJ Open]

ARTICLE DETAILS

TITLE (PROVISIONAL)	Short-term outcomes after emergency surgery for complicated peptic ulcer disease from the UK National Emergency Laparotomy Audit: a cohort study.
AUTHORS	Byrne, Benjamin; Bassett, Michael; rogers, chris; Anderson, Iain; Beckingham, Ian; Blazeby, Jane

VERSION 1 – REVIEW

REVIEWER	Kjetil Søreide University of Edinburgh
REVIEW RETURNED	14-May-2018

GENERAL COMMENTS	This is a well-presented paper and gives an overview of emergency surgery for complicated ulcer disease in the NELA cohort. The results are of interest for the group "complicated ulcer" as such, but I think the cohort largely mixes apples and oranges. It should be made clear that surgery is the treatment of choice and applied to the vast majority of perforations, but in appropriately managed bleeding ulcers, surgery is performed in <5% of cases due to advances in endoscopy, adjunct techniques and intervention radiology. So, the bleeding ulcers that proceed to surgery are a select group (should be) that are often in extremis, compared to the perforated cases. Have you explored if units provide 24/7 endoscopy, interventional radiology, CT imaging, ICU/HDU beds etc? I am unsure if the figure on heart-rate and mortality provides any useful information, particularly for bleeders that may also present initially as "occult" bleeding and be worked up over the next 24-48 hours and slowly but surely deteriorating with several attempts at endoscopy, embolisation, transfusions, etc. Prognostic models for perforations have shown only modest ability to predict outcome, so this may be discussed to some degree as well. Issues around this have been highlighted in number of papers of the recent past years, see for example: Lu Y, Loffroy R, Lau JY, Barkun A. Multidisciplinary management strategies for acute non-variceal upper gastrointestinal bleeding. Br J Surg. 2014 Jan;101(1):e34-50. doi: 10.1002/bjs.9351. Epub 2013 Nov 26. Review. PubMed PMID: 24277160. Jairath V, Kahan BC, Logan RF, Hearnshaw SA, Dore CJ, Travis SP, Murphy MF,
---

	Palmer KR. National audit of the use of surgery and radiological embolization after failed endoscopic haemostasis for non-variceal upper gastrointestinal bleeding. Br J Surg. 2012 Dec;99(12):1672-80. Lanas A, Chan FKL. Peptic ulcer disease. Lancet. 2017 Aug 5;390(10094):613-624. doi: 10.1016/S0140-6736(16)32404-7. Søreide K, Thorsen K, Harrison EM, Bingener J, Møller MH, Ohene-Yeboah M, Søreide JA. Perforated peptic ulcer. Lancet. 2015 Sep 26;386(10000):1288-1298.
--	---

REVIEWER	Itamar Ashkenazi Hillel Yaffe Medical Center, Hadera, Israel
REVIEW RETURNED	02-Jun-2018

GENERAL COMMENTS	Title: Short-term outcomes after emergency surgery for complicated peptic ulcer disease from the National Emergency Laparotomy Audit (bmjopen-2018-023721) Methods: Cohort of patients recorded in a national registry Authors' methods and main findings: The National Emergency Laparotomy Audit (NELA) is a mandatory audit of emergency bowel operations in England and Wales established in 2012. Of 43,321 emergency laparotomies registered, data was analyzed on 2444 (5.5%) procedures for perforated ulcers and 382 (0.9%) bleeding ulcers. Age was 57.8 and 65.0 years in patients with perforation and bleeding respectively. The proportion of patients with ASA 4-5 was 21.6% and 49.8% in patients with perforation and bleeding respectively. Predicted mortalities and morbidities were higher in bleeding patients. Median time to operation was 8.8 hours and 30.4 hours in patients with perforation and bleeding respectively. Most of this time was taken up by workup and treatment up until a decision was taken to operate the patients. Once a decision was made to operate, it took only 1.1-2 hours to operate these patients. Preoperative CT was done in 74.1% and 26.8% patients with perforation and bleeding respectively. Compared to patients with perforation, more patients with bleeding were admitted postoperatively to high dependency units or intensive therapy units (58.3% vs 81.4%). Overall 60-day in-hospital mortality was 11.7% in patients with perforation, and 17.8% in patients with bleeding. Age, high ASA, low preoperative SBP, and higher post-operative level of care were associated with mortality. Gender, preoperative HR, preoperative CT, time from admission to operation, operating surgeon, operative approach, intraoperative contamination were not associated with mortality. Authors' main conclusions: Mortality after repair of perforated ulcer was primarily associated with patient factors, rather than more easily modifiable aspects of the care provided. This may make it difficult to reduce mortality rates further. This study identified aspects of care that were not associated with mortality suggesting that these should
--

not be the primary focus for immediate quality improvement. The results provide no evidence that more rapid transfer to the operating theatre, greater consultant presence or adoption of minimal access techniques would improve survival. Preoperative SBP is potentially modifiable and may be an appropriate target for future research and quality improvement.

Reviewer's comments: This study is based on a large cohort of patients operated urgently for peptic ulcer complications. Once a very common indication for surgery, due to improvements in treatment, the number of operations for peptic ulcer has diminished considerably compared to >3 decades ago. The number of studies describing patients treated during the last 1-2 decades is small. This study, therefore, is one more addition to the relatively limited literature on this topic concerning patients treated in recent years. Similar to the other studies, this study shows considerably high postoperative mortality, especially in patients operated upon for bleeding. Due to the relatively limited number of recent studies on this topic and the high mortality, this paper deserves to be published. There are however some issues that need to be addressed:

1. The most important finding is that mortality is high in these patients, more so in bleeding patients. This has been published. This manuscript could offer some insight to mortality factors that may be reversible. Factors assessing mortality were done for patients with perforation together with those with bleeding. However, these two groups are not similar. Most if not all patients with perforation were operated. This is not true in patients with bleeding, most of whom were not operated. Patients with perforation are operated on soon after diagnosis. Patients with bleeding are operated only after 1 or 2 gastroscopies (and maybe angiography) have failed. These are different patients that should not be analyzed together.
2. The authors present data that needs to be explained:
 - a. Why were patients with perforation operated upon only 8.8 (5.3-18.9) hours following admission? Is this common?
 - b. Why did 26.8% patients with bleeding undergo preoperative CT?
 - c. Why did patients with bleeding receive antibiotics? Is that part of the protocol?
3. Data on average institutional surgical volume for bleeding ulcers was presented in the abstract. I did not find this data in the results section.
4. The authors found that there is no association between time from admission to operation. They interpret this finding as suggesting that appropriate investigations or fluid optimization is unlikely to compromise survival outcomes. They even suggest that since timing to operation is not associated with mortality, improving patients' physiological condition could improve SBP, a factor which was associated with outcome in this study. Though the authors caution us

regarding this conclusion, it should be said that this study was not constructed to answer this question. There is no data presented to suggest that the patients in the first quartile were similar to those in other quartiles. There is not data presented to allow us to conclude that patients operated late received more treatment in order to improve their physiological condition.

5. Some of the data in the tables and figure one are not clear:
 - a. Table 1 – Age in years for perforation is 57.8 (19.4)... My guess is that 19.4 is standard deviation. Why should the reader be guessing? The heading % is misleading. The authors should choose median and interquartile ranges since most of the continuous data is presented that way and try to find a way that the reader understands what is the data that is being presented.
 - b. Table 4 – same comment, but then, this entire table is unnecessary since the data appears within the text anyways.
 - c. Figure 1 – what does mortality risk minus 2 or minus 3 mean? The authors explain in the text that extremes of SBP were associated with higher mortality. Which SBP was the standard?
6. The main limitation of this study is that it is based on a registry. Registries have the benefit of containing information on many patients. Other than that, one needs to be cautious when interpreting registry data. Even though data is collected prospectively, the data contained in the registry does not always address the question being asked. In this regard, retrospective studies may have an advantage over registries since the data extracted from the files is oriented towards the question being asked. For example:
 - a. The patients' SBP proved to be a risk factor for adverse outcome. The authors reached a conclusion that SBP should be corrected since timing to surgery does not influence survival. I ask the authors, what is the significance of SBP which is measured just before surgery? If SBP is normal, does it prove that patients were optimally resuscitated before surgery? Maybe these patients were healthier from the start?
 - b. The authors acknowledge that some of the patients were probably not even registered, such as patients with comorbidities that were treated nonoperatively. The authors also ask themselves whether some sites may have not registered patients who underwent laparoscopic surgery. Indeed, inclusion criteria or misunderstanding of these criteria may exclude some pertinent patients and lead to selection bias.
 - c. The data's quality within the registry may be impaired. The authors question what is the value of

	the information concerning contamination in patients undergoing surgery for bleeding. I ask the same question. This shows that the definition for this variable was not clarified enough. This may be true with other variables in this registry. The assumption described above that some of the sites probably did not register patients undergoing laparoscopy only strengthens this observation. All these highlight possible problems in studies done on registry data.
--	--

VERSION 1 – AUTHOR RESPONSE

Response to reviewers

This is a well-presented paper and gives an overview of emergency surgery for complicated ulcer disease in the NELA cohort.

The results are of interest for the group "complicated ulcer" as such, but I think the cohort largely mixes apples and oranges. It should be made clear that surgery is the treatment of choice and applied to the vast majority of perforations, but in appropriately managed bleeding ulcers, surgery is performed in <5% of cases due to advances in endoscopy, adjunct techniques and intervention radiology. So, the bleeding ulcers that proceed to surgery are a select group (should be) that are often in extremis, compared to the perforated cases.

We thank the reviewer for highlighting this issue. We have further underlined the different roles of surgical treatment for perforation and bleeding ulcers in the first paragraph of the introduction. Paragraph 6 of the discussion also highlights the secondary role of surgery for bleeding ulcers, after endoscopic and/or radiological treatment.

Have you explored if units provide 24/7 endoscopy, interventional radiology, CT imaging, ICU/HDU beds etc?

Data on availability of these supporting services is not available in NELA. Most acute hospitals will have 24/7 endoscopy for bleeding patients, CT imaging, ICU and HDU. Radiology is less widely available. We have added a statement about its availability with a citation to paragraph 6 of the discussion. However, unit-level data is not available for modelling within this study.

I am unsure if the figure on heart-rate and mortality provides any useful information, particularly for bleeders that may also present initially as "occult" bleeding and be worked up over the next 24-48

hours and slowly but surely deteriorating with several attempts at endoscopy, embolisation, transfusions, etc.

This figure illustrates the shape of the relationship between pre-operative systolic blood pressure and risk of mortality for patients undergoing surgery for perforation only. It does not include patients undergoing surgery for bleeding peptic ulcer disease. This has been clarified in the legend for this figure.

Prognostic models for perforations have shown only modest ability to predict outcome, so this may be discussed to some degree as well.

Issues around this have been highlighted in number of papers of the recent past years, see for example:

Lu Y, Loffroy R, Lau JY, Barkun A. Multidisciplinary management strategies for acute non-variceal upper gastrointestinal bleeding. *Br J Surg*. 2014

Jan;101(1):e34-50. doi: 10.1002/bjs.9351. Epub 2013 Nov 26. Review. PubMed PMID: 24277160.

Jairath V, Kahan BC, Logan RF, Hearnshaw SA, Dore CJ, Travis SP, Murphy MF, Palmer KR. National audit of the use of surgery and radiological embolization after failed endoscopic haemostasis for non-variceal upper gastrointestinal bleeding. *Br J Surg*. 2012 Dec;99(12):1672-80.

Lanas A, Chan FKL. Peptic ulcer disease. *Lancet*. 2017 Aug 5;390(10094):613-624. doi: 10.1016/S0140-6736(16)32404-7.

Søreide K, Thorsen K, Harrison EM, Bingener J, Møller MH, Ohene-Yeboah M, Søreide JA. Perforated peptic ulcer. *Lancet*. 2015 Sep 26;386(10000):1288-1298.

These papers are interesting, and we thank the reviewer for the citations. We have added a brief discussion of risk prediction models in paragraph 3 of the discussion, highlighting the difficulties, and signposting our focus on assessing current practice and care provision, rather than trying to create a new risk prediction model for individual patient decision-making.

Reviewer 2

Reviewer's comments: This study is based on a large cohort of patients operated urgently for peptic ulcer complications. Once a very common indication for surgery, due to improvements in treatment, the number of operations for peptic ulcer has diminished considerably compared to >3 decades ago. The number of studies describing patients treated during the last 1-2 decades is small. This study, therefore, is one more addition to the relatively limited literature on this topic concerning patients treated in recent years. Similar to the other studies, this study shows considerably high postoperative mortality, especially in patients operated upon for bleeding. Due to the relatively limited number of recent studies on this topic and the high mortality, this paper deserves to be published. There are however some issues that need to be addressed:

1. The most important finding is that mortality is high in these patients, more so in bleeding patients. This has been published. This manuscript could offer some insight to mortality factors that may be reversible. Factors assessing mortality were done for patients with perforation together with those with bleeding. However, these two groups are not similar. Most if not all patients with perforation were operated. This is not true in patients with bleeding, most of whom were not operated. Patients with perforation are operated on soon after diagnosis. Patients with bleeding are operated only after 1 or 2 gastroscopies (and maybe angiography) have failed. These are different patients that should not be analyzed together.

We have revised the manuscript to make it clearer that we have not analysed the two patient groups together. We have highlighted in the first paragraph of the introduction that the role of surgery differs for perforated vs bleeding PUD.

2. The authors present data that needs to be explained:

- a. Why were patients with perforation operated upon only 8.8 (5.3-18.9) hours following admission? Is this common?

The data presented in table 2 give some insights into reasons for this time interval. The majority of the interval from admission to operation is taken up in time from admission to decision to operate, at a median of 6.0 hours. This interval will be composed of the initial admission process, subsequent senior review, and blood and/or radiological testing. While further details on this process are not available within the NELA data, we think this is plausible and appropriate for the full work-up and decision-making process, the individual process of which will be familiar to clinicians managing such patients. The current manuscript is just over 3000 words long and we have therefore omitted detailed discussion of this finding.

b. Why did 26.8% patients with bleeding undergo preoperative CT?

The NELA dataset does not allow specification of types of CT imaging, therefore pre-operative CT angiography will have been classified as pre-operative CT. We have added a sentence to paragraph 6 of the discussion about this high use of CT prior to surgery for bleeding, the lack of specific detail on the type of CT, and that NELA may need to capture this information in the future.

c. Why did patients with bleeding receive antibiotics? Is that part of the protocol?

273 of 382 bleeding patients received pre-operative antibiotics. We cannot define the reasons for this from the data available. However, national guidance in the UK advises that prophylactic antibiotics are indicated for suspected or confirmed variceal bleeding. Therefore, patients may have been given antibiotics prior to their first endoscopy before a final diagnosis of bleeding secondary to PUD (rather than varices).

3. Data on average institutional surgical volume for bleeding ulcers was presented in the abstract. I did not find this data in the results section.

Figures for median (interquartile range) institutional volume for perforation and bleeding is presented at the end of the first paragraph of the results, under the heading 'Patient characteristics and hospital volume.'

4. The authors found that there is no association between time from admission to operation. They interpret this finding as suggesting that appropriate investigations or fluid optimization is unlikely to compromise survival outcomes. They even suggest that since timing to operation is not associated with mortality, improving patients'

physiological condition could improve SBP, a factor which was associated with outcome in this study. Though the authors caution us regarding this conclusion, it should be said that this study was not constructed to answer this question. There is no data presented to suggest that the patients in the first quartile were similar to those in other quartiles. There is not data presented to allow us to conclude that patients operated late received more treatment in order to improve their physiological condition.

As well as highlighting the exploratory nature of our multivariate analysis throughout the manuscript, we have revised paragraph 3 of the discussion to make our cautious interpretation of this finding clearer. We have also more clearly signposted that this should be the subject of further research to evaluate the role of pre-operative optimisation.

5. Some of the data in the tables and figure one are not clear:

a. Table 1 – Age in years for perforation is 57.8 (19.4)... My guess is that 19.4 is standard deviation. Why should the reader be guessing? The heading % is misleading. The authors should choose median and interquartile ranges since most of the continuous data is presented that way and try to find a way that the reader understands what is the data that is being presented.

The % in the column heading reflects the data for the majority of rows in the table. We agree the reader should not have to guess. Where the data do not reflect n and % we have made this clear in the row label, e.g. age in years (mean (SD)), X (median (IQR)). The choice of a mean over median (or vice versa) was decided on the basis of the distribution of the data. The mean/median reflects the average value and the SD/IQR the spread. For an approximately normally distributed variable the mean and SD are the most appropriate measures to use, the median and IQR are used when the distribution is skewed.

b. Table 4 – same comment, but then, this entire table is unnecessary since the data appears within the text anyways.

Table 4 has been edited to make the data clearer. We acknowledge that most of the data is also presented in the text, but find that a summary table is useful for readers to be able to appraise and compare results.

c. Figure 1 – what does mortality risk minus 2 or minus 3 mean? The authors

explain in the text that extremes of SBP were associated with higher mortality. Which SBP was the standard?

This figure is included only for illustration purposes, and axis labels have now been removed. We considered that most readers would find it easier to understand the distribution of the association between systolic blood pressure and mortality with a figure, rather than through prose.

6. The main limitation of this study is that it is based on a registry. Registries have the benefit of containing information on many patients. Other than that, one needs to be cautious when interpreting registry data. Even though data is collected prospectively, the data contained in the registry does not always address the question being asked. In this regard, retrospective studies may have an advantage over registries since the data extracted from the files is oriented towards the question being asked. For example:

a. The patients' SBP proved to be a risk factor for adverse outcome. The authors reached a conclusion that SBP should be corrected since timing to surgery does not influence survival. I ask the authors, what is the significance of SBP which is measured just before surgery? If SBP is normal, does it prove that patients were optimally resuscitated before surgery? Maybe these patients were healthier from the start?

Thank you for this helpful comment. As discussed above, we have edited paragraph 3 of the discussion to discuss this finding more cautiously, proposing further research to evaluate the role of pre-operative optimisation.

b. The authors acknowledge that some of the patients were probably not even registered, such as patients with comorbidities that were treated nonoperatively. The authors also ask themselves whether some sites may have not registered patients who underwent laparoscopic surgery. Indeed, inclusion criteria or misunderstanding of these criteria may exclude some pertinent patients and lead to selection bias.

We have added a sentence in the limitations section highlighting that those with complicated PUD who were successfully managed without surgery were not included.

c. The data's quality within the registry may be impaired. The authors question what is the value of the information concerning contamination in patients undergoing surgery for bleeding. I ask the same question. This shows that the definition for this variable was not clarified enough. This may be true with other variables in this registry. The assumption described above that some of the sites probably did not register patients undergoing laparoscopy only strengthens this observation.

All these highlight possible problems in studies done on registry data.

We agree it is important to highlight the problems with registry data and discuss the many limitations of this study in the penultimate paragraph of the discussion. This includes potential problems with case ascertainment, selection bias, coding accuracy, missing data and the selected outcome. However, NELA represents the most clinically detailed dataset for study of this patient population, and should be used, with appropriate caution, to examine current practice and direct future research.

VERSION 2 – REVIEW

REVIEWER	Itamar Ashkenazi Hillel Yaffe Medical Center, Israel
REVIEW RETURNED	07-Jul-2018
GENERAL COMMENTS	The authors clarified my questions and made the appropriate changes where needed. Thank you for allowing me to review the revision.